# Biochemical Characterization and Structural Insight into Interaction and Conformation Mechanisms of *Serratia marcescens* Lysine Decarboxylase (SmcadA)

**DOI:** 10.3390/molecules26030697

**Published:** 2021-01-29

**Authors:** Tolbert Osire, Zhina Qiao, Taowei Yang, Meijuan Xu, Xian Zhang, Zhiming Rao

**Affiliations:** The Key Laboratory of Industrial Biotechnology, Ministry of Education, School of Biotechnology, Jiangnan University, 1800 LiHu Boulevard, Wuxi 214122, China; tobykrane@gmail.com (T.O.); 6170206026@stu.jiangnan.edu.cn (Z.Q.); xumeijuan@jiangnan.edu.cn (M.X.); zxshengwu@126.com (X.Z.)

**Keywords:** lysine decarboxylase, *Serratia marcesenes*, structural conformation, cofactor, interactions

## Abstract

Inducible lysine decarboxylases (LDCs) are essential in various cellular processes of microorganisms and plants, especially under acid stress, which induces the expression of genes encoding LDCs. In this study, a novel *Serratia marcesenes* LDC (SmcadA) was successfully expressed in *E. coli*, purified and characterized. The protein had an optimal pH of 6 and a temperature of 40 °C and phylogenetic analysis to determine the evolution of SmcadA, which revealed a close relation to *Enterobacteriaceae, Klebsiella* sp., among others. The molecular weight of SmcadA was approximately 75 kDa after observation on SDS-PAGE and structural modeling showed the protein as a decamer, comprised of five interlinked dimers. The biocatalytic activity of the purified wild-type SmcadA (WT) was improved through site directed mutations and the results showed that the Arg595Lys mutant had the highest specific activity of 286.55 U/mg, while the Ser512Ala variant and wild-type SmcadA had 215.72 and 179.01 U/mg, respectively. Furthermore, molecular dynamics simulations revealed that interactions through hydrogen bonds between the protein residues and cofactor pyridoxal-5-phosphate (PLP) are vital for biocatalysis. Molecular Dynamics (MD) simulations also indicated that mutations conferred structural changes on protein residues and PLP hence altered the interacting residues with the cofactor, subsequently influencing substrate bioconversion. Moreover, the temperature also induced changes in orientation of cofactor PLP and amino acid residues. This work therefore demonstrates the successful expression and characterization of the purified novel lysine decarboxylase from *Serratia marcesenes* and provided insight into the mechanism of protein–cofactor interactions, highlighting the role of protein–ligand interactions in altering cofactor and binding site residue conformations, thus contributing to improved biocatalysis.

## 1. Introduction

Inducible amino acid decarboxylases have been implicated in various cellular processes in microorganisms and plants [1], for example, they regulate acid induced stress in bacterium existing in the stomach and urinary tract of microorganisms. The buffering effect to acid stress is achieved by the bioconversion of amino acids (lysine, ornithine) to polymines such as cadaverine putrescine, and spermidine, which are highly alkaline and thus neutralize pH [2,3]. Polyamines also play a significant role in shielding microbial cells against superoxide stress through the formation of siderophores, small molecules that are crucial for iron sequestration/scavenging and are vital as plant/animal defense systems for the manifestation of complete virulence [4]. Furthermore, polyamines are engaged in modulating the synthesis of DNA and RNA [5], and sometimes make up the outer membrane of Gram-negative bacteria [6], thus underlying the diverse role of amino acid decarboxylases.

Indeed, microbial lysine decarboxylases (LDCs) have been extensively studied and have been categorized into two distinct groups: the constitutive LDCs, which are essential in cellular metabolic processes and pH sensitive inducible LDCs involved in the conversion of lysine to cadaverine [7], a polyamine which modulates cellular pH, and is associated with the adaptation of microorganisms like *Escherichia coli*, Vibrio cholerae, and Salmonella enterica to acidic conditions, forming part of outer cell membranes of Gram-negative bacteria by interacting with peptidoglycan. It is thus necessary for membrane integrity [8,9] and is required for biofilm formation [10]. Recently, there is increased research interest in LDCs [11,12], owing to the vast potential applications of cadaverine as a ‘green’ alternative building block in the synthesis of bio-polyamides [13], which are required for the production of fungicides, pharmaceuticals, fabric softeners, biodegradable plastics, and additives among others [14].

Moreover, LDCs and ornithine/arginine decarboxylase rely on pyridoxal-5-phosphate (PLP) as a cofactor for their activity [3,15]. The significance of PLP-dependent enzyme reactions is vast and the specificity diverse hence increased interest in the protein structure-function as well as protein-cofactor interaction studies. Despite many studies on the physiological function of LDCs and advancement of protein structure technology, the underlying mechanism of protein-ligand interaction influence on decarboxylation is still under studied. Furthermore, studies have reported intramolecular signal transmission in proteins through a network of covalently and non-covalently bonded residue interactions [16], where conformational changes could be induced by point mutations, binding with ligands or changes in pH, ions, substrate concentration, among other factors [17].

Therefore, in this study, we expressed, purified and characterized the *Serratia marcescens* lysine decarboxylase (SmcadA) in *E. coli.* Coupled with increased interest in understanding the mechanisms of protein–protein, protein–ligand interactions, and allosteric orientations, we applied molecular dynamics simulation to gain insight into the structure–functional mechanism of protein–cofactor interactions for the wild-type and two site-directed mutations (Ser512Ala and Arg595Lys) constructed and characterized in our related study [18].

## 2. Results

### 2.1. Phylogeny, Residue Conservation Profiling and Sequence Analysis

To identify the evolutionary differences of the LDCs in different species with respect to SmcadA, we searched for homogenous sequences on the PSI-BLAST (Position-Specific Iterated BLAST) database at an E-value below 5 × 10^−4^, and multiple sequence alignments of the top 1000 related proteins obtained were used for the construction of the phylogenetic tree and residue conservation profiling. The Phylogenetic tree revealed that the *Serratia marcesenes* lysine decarboxylase (SmcadA) had a high similarity to LDCs from *Enterobacteriaceae, Klebsiella* sp., *pantoea deleyi*, *Kluyvera ascorbate, Salmonella* sp., *Raoultella* sp., *Escherichia coli* sp., and *Metakosakonia massiliensis* (Figure 1A). this means that LDCs from the above strains evolved from the same ancestry and thus could be involved in related cellular processes.

Analysis of the protein sequence on the WebLogo webserver showed that the protein was highly conserved, which was in agreement with a previous study that revealed high evolutional conservation of LDC residues, which also justifies the compact nature of the structure of LDCs compared to ornithine/arginine decarboxylases [19]. In our related study [18], Ser512Ala and Arg595Lys point mutations significantly increased the biotransformation of L-lysine to cadaverine. Therefore, we performed residue profiling of protein residues on WebLogo, and evidently, the results showed that the Ser512 residue (corresponding to Ser514) and Lys595 (corresponding to residue 597), both residues marked with red star, and were highly conserved in the majority of LDCs (Figure 1B), suggesting that these residues played a key role in the functioning of this enzyme.

### 2.2. Protein Structure, Expression and Purification

The modeled SmcadA protein structure was a homo decamer comprised of five interlinked dimers, forming five pairs of symmetrically parallel rings, with a 91.41% similarity to inducible lysine decarboxylase (PDB ID: 3q16.1A). The cofactor PLP binding pocked was predicted to be located at the dimer interface buried within the regions 415–483 (consisting of loops 415–417; 445–483 and α-helices 418–444; 543–561) of the first unit and adjacent region 83–109, made of loop regions 83–90; 94–99; 104–108, α-helix 91–93 and β-sheet 100–103 of the second monomer unit. Existing literature iterated that dimerization is crucial for enzyme activity in LDCs by controlling the conformation of the active site residues within the dimer interface [20], and seemingly, the loops 83–91; 94–99 together with β-sheet 100-103 formed a lid at the entrance of the PLP binding pocket.

Conforming to related LDC structures [21], each monomer was made of three domains: the wing domain having residues 1–128 formed the N-terminal, the core domain consisted of residues 129–562, while the C-terminal domain (CTD), predominantly consisting of α-helix region 563–712 (Figure 2A). Indeed, the SDS page analysis of the expressed protein in *E. coli* recombinant strain BL21/pET28-SmcadA revealed its molecular weight as approximately 75 kDa (Figure 2B). To further validate the structure and size of SmcadA, native PAGE and western blot gel electrophoresis were performed, and as shown in Appendix A, in Appendix A, SmcadA had a higher molecular weight compared to Bovine Serine Albumin (66.4 kDa) and Proteinase K (28.9 kDa), and further showed that the protein was partially charged, as evidenced by its slow movement in basic buffer.

### 2.3. Characterization of Purified SmcadA and Variants

Decarboxylses have been implicated in a wide range of cellular processes and their mechanism of activity is diverse. Moreover, lysine decarboxylases exhibit allosteric properties in response to pH, temperature, substrate and cofactor concentrations [22,23], resulting in interactions between protein residues, substrate and ligands or cofactors [24,25]. Therefore, we characterized the optimal conditions for the efficient conversion of L-lysine to cadaverine by pure SmcadA and variants. The results showed that SmcadA and variants were active over a pH range of 6–9 with optimal conversion at pH 6. The optimum temperature for the highest conversion was 40 °C for the WT SmcadA and variants. Furthermore, the results showed that the optimal concentrations of the substrate and cofactor PLP for effective biocatalysis with purified SmcadA were 100 mM and 0.25 mM, respectively (Figure 3A–D).

Moreover, the thermostability results indicated that SmcadA was relatively stable after incubation at 40 °C with a half-life of 7.5 h. Notably, the half-life of the Ser512Ala and Arg595Lys mutants significantly increased to approximately 9 h and 8.45 h, respectively. However, at high temperatures (60 °C), SmcadA (WT), Ser512Ala and Arg595Lys mutants were highly unstable, possessing half-lives of only 55, 68 and 51 min, respectively (Table 1).

### 2.4. Elucidating the Role of Mutations in Influencing Protein-Cofactor Interactions of SmcadA

Crystal structures combined with protein structure analysis techniques and molecular dynamics simulations exquisitely facilitate the gaining of insight into structural mechanisms of how PLP-dependent enzymes exploit interactions with the cofactor to induce faster conversion rates of specific reactions [15]. Since lysine decarboxylases belong to Fold III type of pyridoxal 5′-phosphate (PLP) dependent decarboxylases [1,26,27], we explored the effect of point mutations on protein–cofactor interactions through MD simulations of the WT and variants in Gromacs.

Our results showed that, the wild-type (WT) and mutants presented differing interactions between the protein residues and the cofactor PLP. For example, the PLP pyridine rings Nʹ and phosphate group of the WT and Ser512Ala mutant formed direct hydrogen bonds with metal ions, while the PLP phosphate groups interacted via hydrogen bonds with Lys543, Asn84 and Ser87, respectively (Figure 4A,B). In the Arg595Lys mutant, the PLP pyridine ring Nʹ formed hydrogen bond with Asn84, while the ring phosphate interacted with the metal ions via hydrogen bond formation (red arrows), as shown in Figure 4C. Furthermore, there were water-mediated interactions between Ala546 and Lys543 (Figure 4C). The high biosynthesis of cadaverine in the Arg595Lys variant could therefore be associated to improved protonation of the substrate as a result of hydrogen bond interactions between PLP and the Lys434 residue [28]. Additionally, the contribution of the hydrogen bonds formed between the pyridine ring Nʹ and Asn84 are known to be crucial for the formation of the carbanionic intermediate and delocalization of its electrons which could attribute to improved cadaverine biosynthesis. Moreover, residue–metal ion interactions also played a significant role in neutralizing the negative charges formed at the transition state during Schiff base formation [29,30]. These results therefore indicated the significant role of point mutations in influencing various protein–cofactor/ metal ion interactions that delocalized electrons and ultimately improved biocatalysis by stabilizing cofactor and intermediates formed. 

### 2.5. Thermal Induced Residue and Cofactor Conformational Rearrangements of SmcadA and Variants

In our thermal stability assays, we observed that biocatalysis with SmcadA and its variants was highly affected by the temperature. Therefore, to gain insight into the mechanism of thermal-induced effects on SmcadA activity, we assessed the structure and conformational changes for the WT and variants through MD simulations in Gromacs at 40 °C and 60 °C. We then aligned the WT and mutants simulation structures at these temperatures and the results showed that, at 40 °C, there was no significant difference in the PLP orientation for the WT and Ser512Ala with the PLP phosphate moiety closely interacting with Thr544 and Leu547, compared to that in the Arg595Lys mutant that closely interacted with Lys434 (Figure 5A).

However, at 60 °C, there was a significant difference in orientations of the cofactor PLP for the WT and its variants, and this contributed to the decreased structural integrity and stability of the protein residues (Figure 5B). The elevated temperature greatly affected the orientation of protein residues in α-helix residues 59–73, loop region 84–95, and C-terminal residue Asp447 (red arrows), altering the cofactor PLP positioning and influencing the biocatalysis. Therefore, the observed PLP conformational changes certainly resulted from temperature induced unfolding of the protein residues causing alterations in residue positions within the binding pocket. These observations support our findings on the effect of temperature on SmcadA and its variants, whereby the WT and Arg595Lys mutant had significantly lower stability at higher temperatures compared to the Ser512Ala mutation (Figure 3B and Table 1).

To further determine the extent of thermal denaturation and unfolding of the individual protein variants, comparative structural analysis of the two SmcadA chains C and E involved in PLP binding for the WT and mutants was performed at 40 °C and 60 °C. The results indicated that, in both the wild-type WT and Ser512Ala mutant, the temperature increase from 40 °C to 60 °C caused significant structural alterations of the protein residues (red arrows) in the N-terminal domain. Specifically, the α-helix residues 59–73 and the loop residues 84–90 (red arrows) were significantly altered in both the WT and Ser512Ala mutant, while there was only a minimal alteration at the C-terminal domain, particularly the Asp447 residue in the WT and none for the Ser512Ala mutant (Figure 6A and Figure 7A). Analysis of the Root mean square deviation (RMSD) for backbone atoms revealed that, at 60 °C, for the WT, there was initially no significant difference in RMSD values within the first 10 ns of the simulation. Afterwards, a decreased RMSD was observed at 60 °C compared to that at 40 °C (Figure 6B), mainly attributed to the increased flexibility of the protein residues, which promoted interactions of cofactor PLP with protein residues and or metal ions. Moreover, the Root Mean Square Fluctuation (RMSF) analysis at 40 °C and 60 °C further showed increased flexibility of the protein residues at high temperature (60 °C) manifested by high RMSF values for unit E N-terminal domain residues and unit C C-terminal domain residues (red rectangles) shown in Figure 6C,D.

Analysis of the backbone RMSD for the Ser512Ala variant showed that the mutant had lower RMSD values at 40 °C (Ser512Ala_40) compared to that at 60 °C (Ser512Ala_60), particularly before 8 ns was significantly lower (Figure 7B). The results further indicated that the RMSF at 60 °C (Ser512Ala_60uE) was significantly higher than that at 40 °C (Ser512Ala_40uE), as shown in Figure 7C by the red rectangle. This suggested that increased temperature caused undesirable unfolding of the protein residues, confirming observed changes in the orientation of residues near Glu65 (Figure 7A). The protein unit C RMSF also showed increased instability of the C-terminal residues (350–600), marked with the red rectangle at 60 °C (Ser512Ala_60uC) compared to that at 40 °C (Ser512Ala_40uC), as shown in Figure 7D. Moreover, these highlighted regions constitute highly conserved residues of the two domains involved in PLP binding. Thus, increased flexibility and instability due to elevated temperatures contributed to the altered interactions of PLP and protein residues at 60 °C [31].

Meanwhile, in the Arg595Lys mutant, unlike in the WT and Ser512Ala mutant, an increase in temperature to 60 °C affected the conformation and structural orientation of both the N-terminal and C-terminal domain residues. Specifically, the N-terminal residues (56–105) and C-terminal domain residues (the α-helix residues 418–444) were significantly altered (Figure 8A). It was also observed that the RMSD and RMSF values at the N and C-terminal for both units E and C were highly affected at 60 °C (Arg595Lys_60uE and Arg595Lys_60uC) compared to 40 °C and at 40 °C (Arg595Lys_40uE and Arg595Lys_40uC), as indicated by their high values (Figure 8B–D). This meant that the integrity of this mutant structure was highly compromised at 60 °C compared to at 40 °C, causing a significant alteration of residue or cofactor PLP positioning, hence affecting biocatalysis [31] and further confirming our results on the temperature effect on biocatalytic activity of SmcadA and its variants.

## 3. Discussion

In response to acidic conditions, microorganisms express lysine decarboxylase genes to convert amino acid lysine to cadaverine, which is known to regulate pH and cellular processes in plants and animals. In this study, we expressed, purified and characterized a novel lysine decarboxylase (SmcadA) from *Serratia marcescens*. Firstly, to determine the evolution process of the lysine decarboxylase from *Serratia marcescens*, phylogenetic analysis revealed that SmcadA had a close relationship to LDCs from *Enterobacteriaceae, Klebsiella* sp., *pantoea deleyi, Kluyvera ascorbate*, *Salmonella* sp., *Raoultella* sp., *Escherichia coli* sp., and *Metakosakonia massiliensis*. Sequence profiling of the SmcadA protein residues on WebLogo revealed a highly conserved protein with 91.41% similarity to *E. coli* inducible lysine decarboxylase and the purified SmcadA protein had a molecular weight of approximately 75 kDa after SDS-PAGE analysis. The SmcadA was found to be active in a wide pH range between 5 and 9, with an optimal pH 6 and temperature of 40 °C. The purified protein was highly active at the substrate and cofactor PLP concentrations of 100 mM and 0.25 mM, respectively. Determination of the decarboxylase activity of the wild-type SmcadA and its variants revealed that the Arg595Lys had significantly enhanced specific activity of 286.55 U/mg, which was 1.6-fold that of the wild-type, while the Ser512Ala variant and WT had 215.72 and 179.01 U/mg, respectively. The SmcadAWT and variants also had apparent Vmax and Km values of 1.076 ± 0.069, 0.934 ± 0.135, 0.963 ± 0.153 and 1.27 ± 0.286, 1.61 ± 0.819, and 2.13 ± 1.039, respectively (Table 2) and the Lineweaver-Burke linear plots for SmcadA WT, Ser512Ala and Arg595Lys mutants, respectively, are shown in Appendix A, in the Appendix A.

MD simulations provided insight into the improved bioconversion of L-lysine to cadaverine. It was evident that mutations altered the overall allosteric integrity of the protein, causing conformational changes in the orientation of the protein residues at the ligand-binding pocket, which ultimately affected key interactions with the cofactor PLP. Moreover, previous studies reported the role of individual residues in PLP-dependent enzyme reactions, for example, hydrogen bond interaction between Asparagine residues (Asn) with phosphate moiety of PLP facilitated carbonionic intermediate and delocalization of electrons of the intermediate. Interactions between metal ions with side chains of protein residues also contributed to increased enzyme catalytic activity, for example, [32] Therefore, the presence of uncharged aromatic amino acids, like Phe102, is known to enhance the proton transfer between Cα and C4ʹ through the neighboring Lys residue [27,33].

Furthermore, we observed thermal induced conformational changes of both PLP and amino acid residues. Particularly, simulations at 60 °C resulted in high RMSD and RMDF values, indicating increased flexibility and the instability of the residues at high temperatures, particularly at the N-terminal. The increased flexibility, although at moderate temperatures, would be advantageous in contributing to the increased interactions between the moieties of cofactor PLP and the protein residues, at high temperatures (60 °C), resulted in improper unfolding of the protein residues, hence affecting biocatalysis. This agreed with our results which indicated reduced biocatalysis at elevated temperatures (Table 1). This study is the first to express, characterize and further demonstrate the evolution through phylogenetic analysis of the *Serratia marcesenes* lysine decarboxylase (SmcadA). This study further applied molecular dynamics simulations to gain insight into the mechanism of various interactions between the cofactor PLP and protein in influencing the biocatalytic activity of the wild-type SmcadA and its mutants, hence providing an alternative approach for the future design of highly efficient biocatalysts.

## 4. Materials and Methods

### 4.1. Materials, Culture Medium and Conditions

All the strains, plasmids and primers used in this study are listed in Table 3. *E. coli* strain BL21 (DE3) was used as a host for plasmid construction and protein expression. The pETDuet-1 plasmid was used as the expression vector and plasmids were constructed by homologous recombination using the ClonExpress II One Step Cloning Kit and transformed into *E. coli* BL21 (DE3) competent cells. All other chemicals used in this study were purchased from Sigma-Aldrich LLC Co. (Shanghai, China). Luria–Bertani (LB) medium consisting of yeast extract (5 g/L), tryptone (10 g/L), and NaCl (10 g/L) at pH 7.4 was used for inoculating the cells used in plasmid construction and protein expression. Single colonies were selected and inoculated into 10 mL of LB media supplemented with 5μg/mL of ampicillin prior to cultivation overnight at 37 °C, 180 rpm. Afterwards, 1% of the overnight cultures was transferred to 50 mL of LB media containing 25 μg/mL ampicillin, then cultured at 37 °C and 180 rpm for 2 h before inducing with isopropyl-β-D-thiogalactoside (IPTG). The induced cultures were then grown at 28 °C and 180 rpm for 16 h. The cells were harvested by centrifugation at 10,000 rpm for 5 min at 4 °C, and the cell pellets were kept at −40 °C and were used for further experiments as whole cells.

### 4.2. Protein Sequence Analysis and Phylogenetic Construction

To understand the genetic relationships between protein homologues from different species of microorganisms, a blast search for non-redundant protein sequences (nr) on the PSI-BLAST (Position-Specific Iterated BLAST) database in NCBI was performed for 1000 sequences at a PSI-BLAST threshold of 5 × 10^−4^. Multiple sequence alignment was done by MUSCLE [34] and the phylogenetic tree generated by the neighbor-joining method in MEGA7 [35] before modification in an online webserver iTOL (http://itol.embl.de/) [36], while the sequence logos, indicating the conserved amino acid residues of the 1000 lysine decarboxylase homologues, were generated online in WebLogo (http://weblogo.berkeley.edu/) [37].

### 4.3. Construction of Recombinant Strains

The recombinant strains were constructed using a modified method by Liu et al. [38]. Briefly, the *Serratia marcescens* W2.3 genome sequence was used to design primers. The lysine decarboxylase coding gene cadA from *Serratia marcescens* was amplified by primers pET28-SmcadA-F and pET28-SmcadA-R, flanked with *Bam*HI *and Hind*III restriction sites, respectively, as listed in Table 3. The resulting amplicon (purified) was ligated to the pET28a expression vector digested previously with the same restriction enzymes, before transformation into *E. coli* BL21 (DE3) competent cells, resulting in the BL21/pET28a-SmcadA recombinant strain. All successful transformations were confirmed by double digestion and sequencing.

### 4.4. Expression and Purification of SmcadA and Variants

The BL21/pET28a-SmcA recombinant was grown at 37 °C, 180 rpm overnight on 10mL LB medium containing 5 μg/mL kanamycin. The overnight cells were then transferred into 50 mL media supplemented with 25 μg/mL kanamycin, cultured in a shaker under the same conditions for 2 h before the addition of IPTG to induce protein expression at 28 °C, 180 rpm. After 16 h incubation, cells were harvested by centrifugation at 8000× *g* rpm, 4 °C for 10 min, then washed with 0.01 M phosphate buffer and sonicated for 15 min, then later centrifuged for 20 min at 10,000× *g* rpm. For the SmcadA variants, the plasmids harboring the mutations were first amplified by PCR from prior recombinants, purified and ligated into pET28a, previously cut with *BamH*I and *Ecor*I, and expressed as described above.

The expressed proteins were purified by Ni2+- affinity chromatography using the AKTA Prime system (GE Healthcare, Sweden) fitted with a His- Trap™ HP column purchased from GE Life Sciences, USA. The crude enzyme (5 mL) was loaded onto the column at 0.5 mL/min with Binding Buffer consisting of 0.02 M Tris–HCl buffer and 0.5 M NaCl, of pH 7.4, and then the enzyme was eluted under a linear gradient of imidazole concentrations from 0–0.5 M at 1 mL/min. Finally, excess imidazole was removed from the purified protein by dialysis using 0.05 M Tris–HCl buffer at pH 7.0 [39]. The purified enzyme fractions were used for activity assay, sodium dodecyl sulfate-polyacrylamide gel electrophoresis (SDS-PAGE) analysis (12% acrylamide) and stored in 10% glycerol at −40 °C for further analysis. To verify that SmcadA is a complex protein (decamer), native PAGE gel electrophoresis was performed, as described by Reference [40]. BSA was used as reference proteins. Additionally, Western blot was performed to ascertain protein expression and molecular weight, as previously described by Uwase et al. [41], with slight modifications.

### 4.5. Biochemical Characterization of the Pure Enzyme and Mutants

The optimum conditions for SmcadA conversion of L-lysine substrate to cadaverine was investigated by carrying out reactions under different temperatures, initial pH, and concentrations of cofactor PLP, and the L-lysine prior to determination of synthesized cadaverine. For thermostability analysis of SmcadA and the variants, the enzyme was first incubated at temperatures between 30–60 °C for 2 h, then the respective residual activities were measured. A modified method by Kikuchi et al. [42] was used to determine the decarboxylase activity of SmcadA and its variants. Briefly, 40 μL of the purified enzymes were incubated at optimal conditions in a reaction mixture containing 500 mM sodium acetate buffer (350 μL), 100 μL substrate L-lysine (100 mM), and 10 μL cofactor PLP (0.25 mM) for 1 h. The reaction was terminated by heating at 100 °C for 5 min. One unit of enzyme activity was defined as the amount of enzyme required to produce 1 µmol of cadaverine per minute at optimal reaction conditions. The kinetic parameters of the purified lysine decarboxylase and its variants were performed as previously described by Han et al. [19] with slight modifications using different lysine substrate concentrations (0.25–10 mM) under the optimal conditions (pH 6.0, 40 °C, in 25 mM sodium acetate buffer). Lineweaver–Burk plots used to calculate kinetic parameters Km and Vmax, according to the enzyme reactions, were generated by Hyper32 software with a non-linear regression Michaelis and Menten model. Protein concentrations were determined using the Bradford method with bovine serum albumin as a standard. All the samples were analyzed in triplicate.

### 4.6. Molecular Dynamics (MD) Simulation and Analysis

The dimer units C and E of the SWISS-MODEL [43] generated decamer protein structure of SmcadA were used for MD simulations on Gromacs [44]. Briefly, the units were prepared for molecular docking by adding hydrogens and energy minimization in UCSF Chimera, then the ‘PDB’ file was saved to the working directory folder. The topology files for the protein and ligand were then obtained and combined to form a complex, which was submerged in a triclinic box of 1.0 nm before energy minimizations. For the respective simulations, two starting trajectories were considered. The first starting trajectory used was the original position of the cofactor PLP after adding it into protein PDB file on Chimera software during Dock preparations, as shown in Appendix A. The protein-cofactor complex was first solvated by executing the command ‘gmx solvate -cp newbox.gro -cs spc216.gro -p topol.top -o solv.gro’, as shown in Appendix A, followed by two energy minimizations prior to NVT equilibration. Once the NVT simulation is complete, we then proceeded with NPT equilibration in order to create the ‘tpr’ files required for MD simulations. Upon completion of the two equilibration phases, the system is then well-equilibrated at the desired temperature and pressure. We finally release the position restraints and run production MD for data collection. It should be noted that the desired temperature adjustments are done by changing the constraints in the ‘npt.mdp’ file prior to the second equilibration (NPT). The protein-ligand MD simulations were done for 20 ns.

To demonstrate that mutations contributed to alteration of the general protein/ligand allosteric conformation, MD simulation trajectory files at 313.15 K for the wild-type WT and mutants (Ser512Ala and Arg595Lys) were manipulated using UCSF Chimera [45], hence interacting residues, active site residue and cofactor PLP conformations determined.

Thereafter, the effects of temperature in inducing conformational rearrangements of active site residues and cofactor PLP in the WT and variants were determined by analyzing trajectory files for simulations performed at 313.15 K and 353.15 K (corresponding to 40 °C and 60 °C), respectively, and then backbone RMSD and residue RMSF values determined to show the thermal effects on structural stability and flexibility.

## Figures and Tables

**Figure 1 molecules-26-00697-f001:**
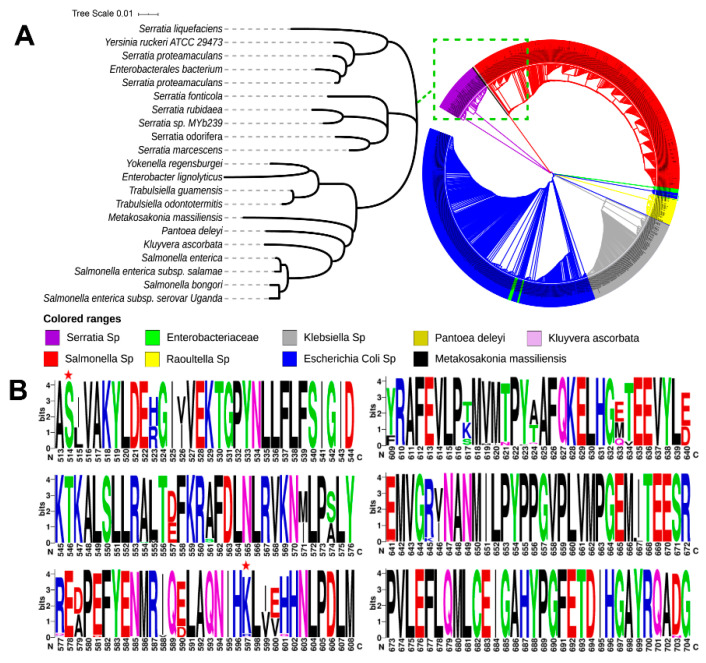
Phylogeny and amino acid residue conservation profiling. (**A**). Phylogenetic tree of the 1000 protein homologues of SmcadA (**B**). Profiling of amino acid residues for conservation using WebLogo. The large, long and wide letter codes represent highly conserved residues.

**Figure 2 molecules-26-00697-f002:**
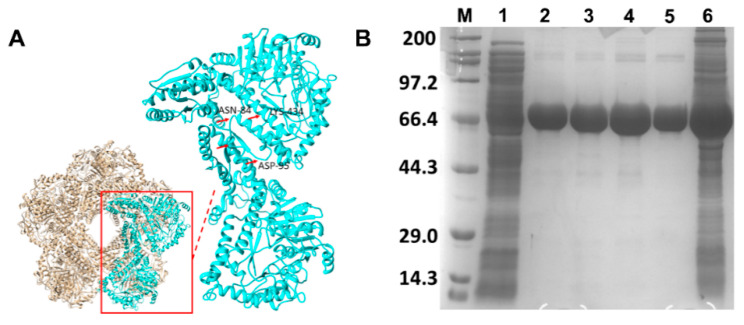
Protein structure and expression. (**A**). The modeled structure of SmcadA, showing the complete decamer and inset, indicates the dimer units with the main residues involved in cofactor PLP binding. (**B**). 12% SDS page image of the expressed proteins. M represented the Protein Marker-Broad (Takara, China); 1 is the *E. coli.* BL21 control strain. 2–5 showed the 75 kDa purified SmcadA. 6 is the unpurified SmcadA.Table 1. This is a table. Tables should be placed in the main text near to the first time they are cited.

**Figure 3 molecules-26-00697-f003:**
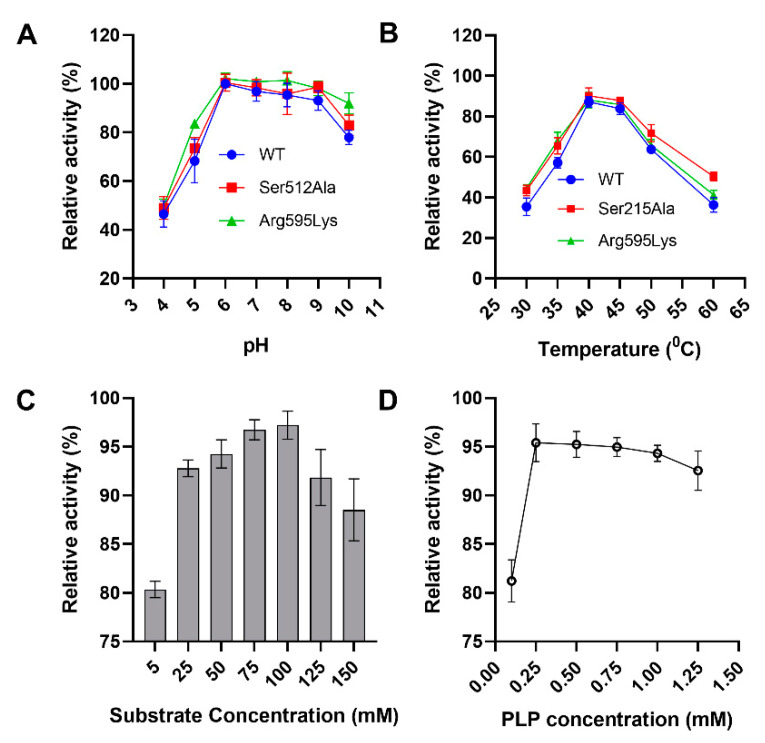
Biochemical characterization of the pure SmcadA. (**A**). A graph showing the optimal pH of SmcadA. The reaction mixtures were incubated at different pHs ranging from 4–10 at 37 °C. (**B**). This shows the optimum temperature after incubating samples at temperature values ranging from 30–60 °C. (**C**) L-lysine substrate concentrations and (**D**). PLP concentrations. The reaction mixture composed of 500 mM acetate buffer at the optimal pH 6.

**Figure 4 molecules-26-00697-f004:**
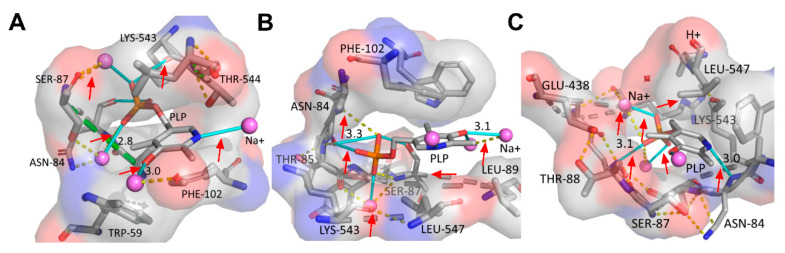
Mutation-induced structural alterations of WT and mutants. (**A**–**C**). This showed the various PLP conformations and interactions between protein residues, cofactor PLP and metal ions for the WT, Ser512Ala and Arg595Lys mutants, at optimum whole-biotransformation temperature of 40 °C. The hydrogen bonds are represented by cyan sticks, the dotted lines (yellow) are polar contacts, while van der Waals interactions are indicated by the green dotted lines.

**Figure 5 molecules-26-00697-f005:**
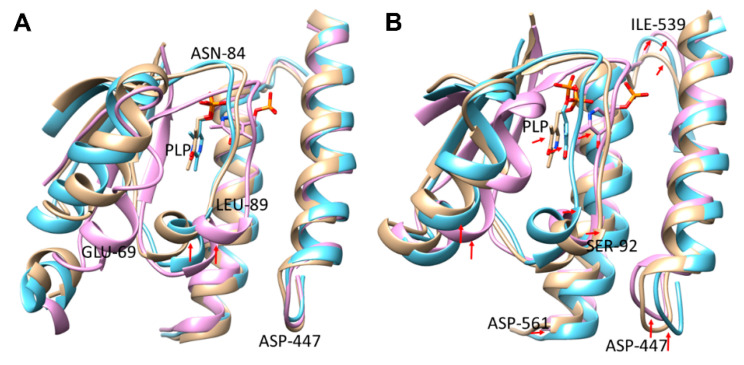
Aligned binding site interacting domains. (**A**) Structure indicating conformational changes of protein residues for the WT and mutants at 40 °C. (**B**). Binding pocket PLP and residue alterations at 60 °C.

**Figure 6 molecules-26-00697-f006:**
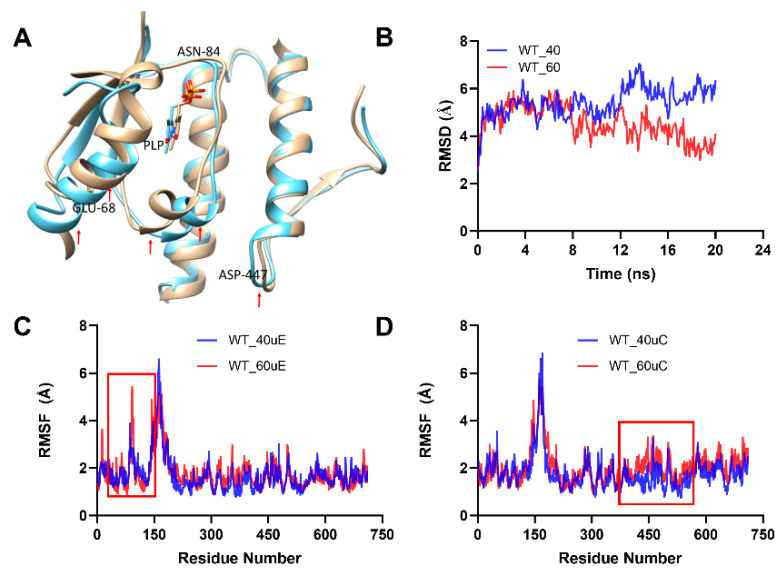
Analysis of thermal-induced structural alterations for the WT. (**A**). This showed the structural variations for the aligned dimer unit involved interactions for the WT at 40 °C and 60 °C, respectively. (**B**). RMSD for the backbone at 40 °C and 60 °C. (**C**,**D**). This showed the RMSF of the two interacting SmcadA units (**C**,**D**) at 40 °C and 60 °C, respectively.

**Figure 7 molecules-26-00697-f007:**
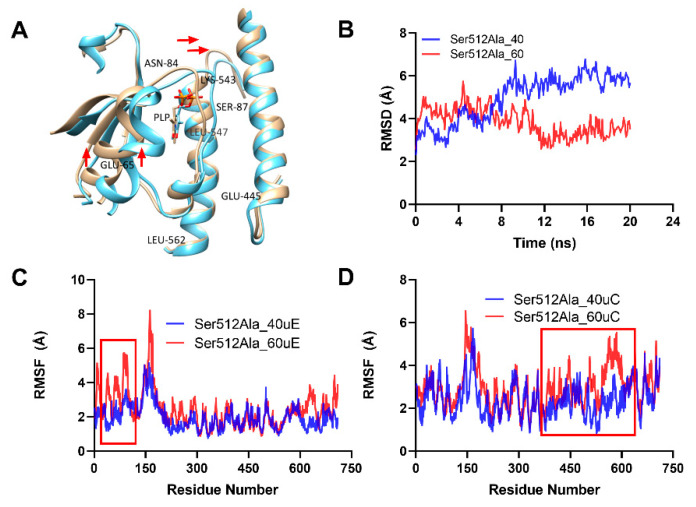
Analysis of thermal-induced structural alterations for the Ser512Ala. (**A**). This showed the structural variations for the aligned dimer unit involved interactions for the WT at 40 °C and 60 °C, respectively. (**B**). RMSD for the backbone at 40 °C and 60 °C. (**C**,**D**). This showed the RMSF of the two interacting Ser512Ala units (**C**,**D**) at 40 °C and 60 °C, respectively.

**Figure 8 molecules-26-00697-f008:**
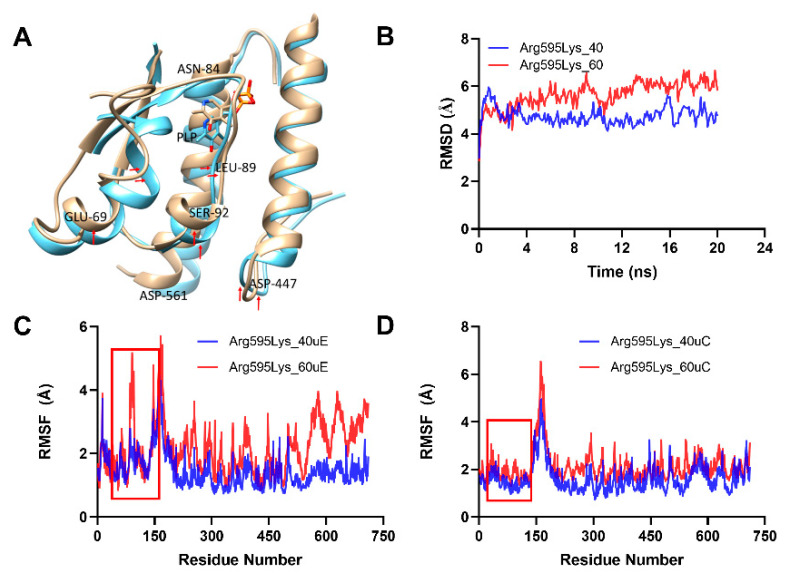
Analysis of thermal-induced structural alterations for the Arg595Lys. (**A**). This showed the structural variations for the aligned dimer unit involved interactions for the Arg595Ly variant at 40 °C and 60 °C, respectively. (**B**). RMSD for the backbone at 40 °C and 60 °C for the. Arg595Ly variant (**C**,**D**). This showed the RMSF of the two interacting Arg595Ly variant units (**C**,**D**) at 40 °C and 60 °C, respectively.

**Table 1 molecules-26-00697-t001:** Thermostability of wild-type SmcadA and mutants.

Enzyme	Tm (°C)	T_1/2_ (min)	Residual Activity (%) after Incubation at 40 and 60 °C for Different Time Intervals (min)	
			40 °C	60 °C
SmcadA WT	40		450	55
Ser512Ala	40		600	68
Arg595Lys	40		558	51

**Table 2 molecules-26-00697-t002:** Specific activity and kinetic parameters of wild-type SmcadA and mutants.

Enzyme	Amount of Protein (mg/mL)	V_max_ (U/mL)	Km (mM)	Specific Activity (U/mg)
SmcadA WT	0.403	1.076 ± 0.069	1.27 ± 0.286	179.01
Ser512Ala	0.353	0.934 ± 0.135	1.61 ± 0.819	215.72
Arg595Lys	0.336	0.963 ± 0.153	2.13 ± 1.039	286.55

**Table 3 molecules-26-00697-t003:** Strains, plasmids and primers used in the study.

Item	Description	Source
Strains		
BL21 (DE3)	*F-dcm ompT hsdS (rB-mB-) gal λ(DE3)*	Laboratory collection
Plasmids		
pMD18T	Cloning vector, 2692 bp, Amp^R^, *lacZ*	TaKaRa
pET28a	*E. coli* expression vector, T7, Amp^R^	Laboratory collection
pET28a-SmcadA	Expression of SmcadA in *E. coli* BL21 (DE3)	This Study
Primers	Primer sequence (5′-3′)	Function
pET28a-SmcadA-F	ccatcatcaccacagcca**ggatcc**atgaacgttatc	Amplification of *S. marcescens* cadA gene
pET28a-SmcadA-R	cttaagcattatgcggccgc**aagctt**ttatttcgccttc	

## Data Availability

All the data is contained within the article and supplementary material. Any further information presented in this study is available on request from the corresponding author.

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
