# Peer review of "Biochemical Characterization and Structural Insight into Interaction and Conformation Mechanisms of Serratia marcescens Lysine Decarboxylase (SmcadA)"

_molecules, 2021, doi:10.3390/molecules26030697_

Round 1

Reviewer 1 Report

The report is important and timely since the literature on this type of enzyme is scarce. Several questions about the results presented with the resources available are for the authors to comment on:

  1. Besides an SDS-PAGE showing protein purity, a native gel or a gel filtration chromatogram is needed to show the quaternary structure and validate the theoretical model presented. This is to support the asseveration that SmcadA is really a decamer.
  2. The authors determined enzymatic activity and the Michaelis-Menten kinetics, but no details are included on how those experiments were made. Please include the Michaelis plots (velocity vs. substrate concentration) and the Lineweaver-Burke linear fit. References and specific details of the decarboxylase assay are required for reference. Important is whether the enzyme obtained the PLP from the bacteria or if it was added before the enzymatic activity assay. Values of Km and Vmax would be much important for experimental comparison between the wild-type and mutant enzymes.
  3. The authors go on a lengthy discussion of the theoretical model as if they have determined a crystal structure, commenting on specific hydrogen bonds that have no experimental evidence they exist. I suggest to tone down the discussion and assert whether the modeling supports the changes in enzymatic activity, and if possible, to the effect on temperature.
  4. Please indicate the identity of SmcadA with the experimental model used as a template for the model.

Reviewer 2 Report

This is a very good paper, which elucidates the molecular mechanism and effect of mutations on the activity of lysine decarboxylase from Serratia marcescens. The Authors found that the Arg595ys mutation exhibited the highest catalytic activity, which they linked to increased conformational changes in substructure and cofactor orientation.

I have the following suggestions and remarks:

1. The analysis of MD results had been carried out correctly but I am not sure how many trajectories were run. If only one per system at a given temperature, this does not seem to be enough and at least one more per system should be run to confirm the trend of the conformational changes. It is enough to start the other trajectories from different initial random velocities. 

2. Simulation conditions should also be described in more detail

a) Were the simulations carried out in explicit or in implicit water. The water model (implicit or explicit) should be defined.

b) If the simulations were carried out in explicit solvent, periodic-box shape and dimensions should be defined.

b) It should be stated if the calculations were run in the NpT or in the NVT regime and what thermostat was used.

3. Overall the English is correct but I am not sure what the following phrase in the Abstract (line 32 in page 2) "mutations coffered structural changes". The verb "to coffer" means, according to the Merriam Webster Dictionary, "to store in a box", which does not seem to be relevant here. Shouldn't it be "mutations caused conformational changes" instead?

Round 2

Reviewer 1 Report

I supplementary data figure S1, the molecular weight marker should be shown on each gel and clearly labeled with a font of sufficient size. The figures included are not publication quality, even if they are supplemental. To estimate the molecular weight the authors should do a regression analysis for the native molecular weight markers. 

Same in figure S2. The legends for the axis are unreadable, and they do not show the units for each axis. If each point represents the average of triplicates, the error bar should be shown to show the variability and reliability of the experiments.

Figure S5 should be expanded to a reasonable size they are extremely small to see any detail.
